# LC–MS/MS Analysis of the Emerging Toxin Pinnatoxin-G and High Levels of Esterified OA Group Toxins in Galician Commercial Mussels

**DOI:** 10.3390/toxins11070394

**Published:** 2019-07-05

**Authors:** Paz Otero, Natalia Miguéns, Inés Rodríguez, Luis M. Botana

**Affiliations:** Departamento de Farmacología, Facultad de Veterinaria, Universidad de Santiago de Compostela, 27002 Lugo, Spain

**Keywords:** lipophilic marine toxin levels, Galician mussels, liquid chromatography–mass spectrometry (LC–MS/MS), new emerging toxins

## Abstract

The occurrence of marine harmful algae is increasing worldwide and, therefore, the accumulation of lipophilic marine toxins from harmful phytoplankton represents a food safety threat in the shellfish industry. Galicia, which is a commercially important EU producer of edible bivalve mollusk have been subjected to recurring cases of mussel farm closures, in the last decades. This work aimed to study the toxic profile of commercial mussels (*Mytilus galloprovincialis*) in order to establish a potential risk when ingested. For this, a total of 41 samples of mussels farmed in 3 Rías (Ares-Sada, Arousa, and Pontevedra) and purchased in 5 local markets were analyzed by liquid chromatography tandem mass spectrometry (LC–MS/MS). Chromatograms showed the presence of okadaic acid (OA), dinophysistoxin-2 (DTX-2), pectenotoxin-2 (PTX-2), azaspiracid-2 (AZA-2), and the emerging toxins 13-desmethyl spirolide C (SPX-13), and pinnatoxin-G (PnTX-G). Quantification of each toxin was determined using their own standard calibration in the range 0.1%–50 ng/mL (R2 > 0.99) and by considering the toxin recovery (62–110%) and the matrix correction (33–211%). Data showed that OA and DTX-2 (especially in the form of esters) are the main risk in Galician mollusks, which was detected in 38 samples (93%) and 3 of them exceeded the legal limit (160 µg/kg), followed by SPX-13 that was detected in 19 samples (46%) in quantities of up to 28.9 µg/kg. Analysis from PTX-2, AZA-2, and PnTX-G showed smaller amounts. Fifteen samples (37%) were positive for PTX-2 (0.7–2.9 µg/kg), 12 samples (29%) for AZA-2 (0.1–1.8 µg/kg), and PnTX-G was detected in 5 mussel samples (12%) (0.4 µg/kg–0.9 µg/kg). This is the first time Galician mollusk was contaminated with PnTX-G. Despite results indicating that this toxin was not a potential risk through the mussel ingestion, it should be considered in the shellfish safety monitoring programs through the LC–MS/MS methods.

## 1. Introduction

Marine phycotoxins are produced by harmful microalgae that bioaccumulate in the marine food chain [1]. They are of growing concern for the exploitation of marine living resources in many coastal areas. Every year there are almost 2000 cases of human intoxication due to the consumption of shellfish or fish with 15% mortality [1]. These toxic compounds constitute a threat to environment and human health and restrict the progress of aquaculture, which is one of the fastest growing food sectors in the world.

Galicia (Spain) is the region with the most intense mussel aquaculture in Europe, producing 267,000 tons in 2017 (last published data) [2]. Geographically, the Galician farm areas (Rías) are coastal inlets formed by the sinking of riverbeds that extend from east to west like small fiords. The Instituto Tecnolóxico para o Control do Medio Mariño de Galicia, INTECMAR, is the Public Agency of Regional Government of Galicia that is responsible for the monitoring of the shellfish production areas and for closing it when mollusk toxicity is over the regulated limit. In Europe, the legislated group of lipophilic marine toxins consists of four different chemical groups—yessotoxins (YTXs), azaspiracids (AZAs), pectenotoxins (PTXs), and okadaic acid (OA) and its derivatives—the dinophysistoxins (DTXs). For these toxins, levels found in shellfish for human consumption must be lower than 3.75 mg eq YTX/kg, 0.16 mg eq AZA, and 0.16 mg eq OA/Kg (for the OA and PTX toxin group). Until 2013, all official toxicity determinations in EU countries were carried out through mouse bioassay, but during 2013 and 2014, progressively more determinations were carried out through LC–MS/MS, according to the legislation [3]. The European legislation states that LC–MS/MS-based methods are the technology recognized as the reference standard for the detection of lipophilic marine toxins from 2014 [3]. Therefore, this technique should be applied as the reference method, following indications agreed by the National Reference Laboratories Network [4]. The Codex Alimentarius International Food Standards also includes the criteria for the determination of lipophilic toxins in bivalves, through chemical methods [5].

Besides the marine toxins described above, the emerging toxins called cyclic imines (CIs) can also be present in European waters. This group comprises spirolides (SPXs), gymnodimines (GYMs), pinnatoxins (PnTXs), and pteriatoxins (PtTXs), and they all are macrocyclic compounds with imine and spiro-linked ether moieties [6,7,8]. There are yet no official methods or regulatory limits for the CI toxin group. Their mode of action is based on the interaction with muscle-type and neuronal nicotinic acetylcholine (ACh) receptors (nAChR), which are the main molecular targets involved in their toxicity [9]. From all CI toxins, SPXs were the major causes for concern in the last two decades, due to their widespread global distribution and their high toxicity after i.p. administration to mice [10]. They were first identified in extracts of the digestive glands of mussels (*Mytilus edulis*) and scallops (*Placopecten magellanicus*) from the Atlantic coast of Nova Scotia (Canada), in the early 1990s [11,12,13] and were detected for the first time in Europe in 2005 [14]. Since then, it has increased the CIs presence, 13-desmethyl spirolide C (SPX-13) being the most extended analogue [15,16,17,18]. In addition, PnTXs initially detected in Japanese shellfish in 1995, are now appearing in the coasts of Europe [17,19,20].

Given the increase in reports of emerging toxins, SPXs and PnTXs, coupled with a high demand of Galician mussels, further investigations into potential consumer risk from phycotoxins is required. This study aimed to investigate the toxin profile of the commercial mussel *Mytilus galloprovincialis* (*M. galloprovincialis*), of different sizes, which were obtained from the markets during December 2018 and January 2019. It also assessed the potential consumer risk linked to their consumption. The mussels were from five brands and belonged to three Galician Rías (Ares-Sada, Arousa, and Pontevedra) (Figure 1). Several studies showed the toxin profile of Galician mussels [15,16,18], however, there were no reports concerning the real risk for mussel consumers.

## 2. Results and Discussion

With the aim of characterizing the lipophilic toxin profiles of shellfish from the Galicia market, a complete analysis of the toxin profiles was carried out using the LC–MS/MS technique. Mussels belonging to 5 commercial brands were purchased and analyzed during December 2018–January 2019. First, we analyzed all lipophilic toxins that are currently legislated in the EU (OA, PTXs, YTXs, and AZAs group toxins) [21] and emerging toxins, such as SPXs (SPX-13, SPX 13,19, and SPX-20) and PnTXs (PnTX-A, B, C, D, E, F, and G). Figure 2A–D shows the toxin profile found in a mussel sample originally from Ría de Pontevedra and obtained in a market from Lugo (Galicia). Results showed that the regulated toxins OA (RT = 4.18 min), DTX-2 (RT = 4.31 min), PTX-2 (RT = 4.38 min), AZA-2 (RT = 4.9 min), and the emerging toxin SPX-13 (RT = 2.75 min) were found in a number of the analyzed samples. This is the typical toxin profile in the Galician coast, where OA and DTX-2 occur due to *Dynophisis acuta* and *Dynophisis acuminata* [22], PTX-2 occur due to *D. acuta* and *Dynophisis caudata* [23], and AZA-2 are produced by *Azadinium* spp. [24]. The presence of SPX-13 is also frequent in Galicia and other European regions, due to the presence of *Alexandrium ostenfeldii* or *Alexandrium peruviaunum* [25,26], and for this reason, this toxin is often included in the monitoring programs [15]. In addition, analyses showed the presence of the emerging toxin pinnatoxin-G (PnTX-G) (RT = 3.1 min) in some samples (Figure 2E). These toxins are produced by *Vulcanodinium rugosum* dinoflagellate [9].

Figure 3 shows the chromatograms of three specific PnTX-G multiple reaction monitoring (MRM) transitions (*m*/*z* 694.5 > 164.1, *m*/*z* 694.5 > 440.1 and *m*/*z* 694.5 > 676.5) in Sample No. 12 and in the standard (1 ng PnTX-G/mL in methanol). Although the intensity of the chromatograms is not very high, we could confirm the presence of PnTX-G in 5 out 41 samples tested, 4 belonged to Ría de Arousa (Sample No. 9, 12, 15, and 31) and one belonged to Ría de Pontevedra (Sample No. 11). The identification was performed through a comparison of the retention time with that of the standard, by means of the specific three transitions and the ratio between these transitions in the samples and the standard. In the last decade, this toxin was recorded in shellfish from Catalonia (Spain) [17,20], Norway [19], France [27], Italy, and Slovenia [20]. However, this toxin has not yet been recorded in the Atlantic Coast of Galicia. Apart from PnTXs, the presence of non-regulated AZAs in Europe is well-documented [28,29]. With the aim of checking if some AZAs that are different from the regulated ones, were present in the commercial samples, a total of 44 analogs were monitored in another experiment. The analysis was performed using the same chromatographic conditions as those used for regulated AZAs, including the specific transitions in the MS method described in the Material and Method section (Method 2). Analysis showed that none of the mentioned AZAs were found in the chromatograms (data not shown). Therefore, the only AZA identified in the commercial samples was AZA-2.

Then, all samples were quantified for the identified toxins. The quantification of each toxin was determined using the external standard calibration in the range of 0.1 to 50 ng/mL. To evaluate the extraction process and the matrix effect, a spiked mussel sample was used for toxin recovery and matrix correction, according to the EU-Harmonised Standard Operating Procedure for determination of Lipophilic marine biotoxins in mollusks through LC–MS/MS [4]. The signal suppression/enhancement (SSE) due to matrix and recoveries calculated as a mean of the three values for each toxin is given in Table 1. High recoveries were obtained for PTX-2 (92%), AZA-2 (98%), SPX-13 (110%), and PnTX-G (90%). High values were also obtained for OA (76%) and DTX-2 (85%) that were extracted in their free form, compared to the recoveries obtained when the hydrolysis extraction process was used. In this case, a recovery of 63% for OA and 62% for DTX-2 was obtained. Results from matrix evaluation showed a suppression of signal for PTX-2 (88%), AZA-2 (33%), SPX-13 (70%), and PnTx-G (64%), since the values were lower than 100%. On the contrary, an increase in the signal was observed for OA (162–211%) and for DTX-2 (160–181%).

Table 2 shows the quantification of lipophilic marine toxins found in all commercial mussels, considering the recovery and the matrix correction. In general, we found lipophilic toxins in 38 out 41 of the samples analyzed. The higher levels were for OA ranging from 3.6 µg/kg (Sample No. 25) to 214 µg/kg (Sample No. 22) and were detected in 38 samples, followed by DTX-2, which was detected in 21 samples in quantities of up to 33.5 µg/kg (Sample No. 22), and SPX-13, which was detected in 18 samples in quantities of up to 28.9 µg/kg (sample No. 18). Analysis from PTX-2, AZA-2, and PnTX-G showed smaller amounts. Fifteen samples were positive for PTX-2 with levels ranging from 0.7 (Sample No. 17) to 2.9 µg/kg (Sample No. 21). AZA-2 was observed in 12 samples (0.1–1.8 µg/kg) and PnTX-G in 5 samples, at very low amounts (0.4–0.9 µg/kg). In general, quantities were below the regulated limit for each toxin group. Nevertheless the sum of micrograms of okadaic acid equivalents per kilogram, together with micrograms of pectenoxin equivalents per kilogram, were above 160 µg/kg in 3 samples (Sample Nos. 10, 17, and 22), particularly, due to the high levels of OA and DTX-2 in the form of esters present in these mussel samples. In many bivalves, OA and DTX-2 were transformed to 7-O-acyl derivatives (DTX-3) by esterification with fatty acids of different carbon chain length. Quite likely, this is the main route for the elimination of these compounds from the bivalves [30,31,32]. For example, Sample No. 22 had 30.2 µg/kg of free OA, 183.4 µg/kg of esterified OA, 10.9 µg/kg of free DTX-2, and 22.6 µg/kg of esterified DTX-2, i.e., a total of 247.6 µg/kg. Therefore, the obtained results (Table 2) show that OA, either free or esterified, continue being the main toxin in Galician mollusks and demonstrating that the esterified amount is considerably higher to that in free form, in accordance with previous reports where amount of toxins in the form of the esters is up to 12 times higher [15,18,22]. It is worth mentioning that these three samples contain high levels of the OA group toxins that come from the same common geographical origin (Ría de Arousa) and sell under the same commercial brand for three weeks in a row. In addition, SPX-13 has also been frequent in the Galician coast since 2006, when they appeared for the first time in mollusks, at levels 13–20 µg/kg [16]. On the contrary, when PTX-2 is present, levels in the mollusks are usually low [18,33]. PTX-2 seems to be transformed to PTX-2 seco-acid (PTX-2sa), which is found in concentrations more than ten-fold those of PTX-2 [32]. Finally, the occurrence of AZA-2 in Galician mollusks was recently reported [24]. In this case, AZA-2 was also the most abundant AZA analog out of three AZAs monitored (AZA-1, AZA-2, and AZA-3) and the maximum level estimated was 3 µg/kg, a value that is comparable to the 1.8 µg/kg AZA-2 observed in the present analysis. Finally, the presence of SPX-13 and PnTX-G were recently found at low levels in commercial mollusks from the Mediterranean sea (Catalonia, Italy, and Slovenia) [20]. SPX-13 was detected in 9.4% of the samples analyzed at levels 26–66 µg/kg and PnTX-G was found in 7.3% of the samples, in lower amounts (0.1–12 µg/kg).

The toxins levels found in the Galician commercial mussels (Table 2) are represented in Figure 4, sorted by toxin, mussel size, and place of collection (Ría). Graphs A, C, E, G, I, and K belong to the large Galician mussels (a total of 22 samples) and B, D, F, H, J, and L represent small ones (a total of 19 samples). Comparing the toxin levels for both sizes, the values were found to be quite similar for PTX-2 (Graphs E and F), AZA-2 (Graphs G and H), SPX-13 (Graphs I and J), and PnTX-G (Graphs K and L). However, it seems that OA (Graphs G and H) and DTX-2 (Graphs I and J) had accumulated in larger amounts in large mussels. For example, the maximum OA level found in small mussels was 128.6 µg/kg (Sample No. 9), yet, 5 large mussel samples were in the range 127.2–214 µg/kg. Additionally, the maximum DTX-2 level in mussels of smaller sizes were 17.4 µg/kg, while 5 samples from the large mussels exceeded this value (up to almost double, 33.5 µg/kg in Sample No. 22). Since the number of samples from each collection place was not similar, a proper comparison between Rías was not possible (7 samples from Ría Ares-Sada, 26 samples from Ría Arousa, and 8 samples from Ría Pontevedra). However, it is worth mentioning that neither AZA-2 nor PnTX-G were found in Ría Ares-Sada and only 2 samples from this Ría showed low SPX-13 levels (2.6 µg/kg Sample No. 19 and 4.4 µg/kg Sample No. 14). On the contrary, SPX-13 was found in higher levels in the remaining two Rías, since it was detected in 6 out of the 8 samples in Ría de Pontevedra, at levels of 5.4 µg/kg (Sample No.11) to 28.9 µg/kg (Sample No. 18), and was detected in Ría Arousa in 11 out 26 samples in the range, 0.6 µg/kg (Sample No. 25) to 16.2 µg/kg (Sample No. 9). PTX-2, OA, and DTX-2 amounts were more homogeneous, although higher levels were always found in mussels from Ría Arousa—2.9 µg/kg PTX-2 in Sample No. 21, and 214 µg/kg OA and 33.5 µg/kg DTX-2 in Sample No. 22.

## 3. Conclusions

The occurrence of lipophilic marine toxins from harmful phytoplankton are of growing concern for the Galicia region, both in terms of food safety and economic approaches. In recent years, there is a growing evidence for the presence of emerging toxins in European waters, leading to the possibility that the consumers of these contaminated products might be affected by these new risks. In this work, the EU-Harmonised SOP for the determination of Lipophilic marine biotoxins in mollusks, through LC–MS/MS, was applied to mussels purchased in 5 local markets that originated from 3 farming areas (Ares-Sada, Arousa and Pontevedra). A total of 38 out 41 samples showed lipophilic toxins, including OA and DTX-2, PTX-2, AZA-2, SPX-13, and PnTX-G. Toxins from the OA group were detected in 93% of the samples (7% exceeded the legal limit of 160 µg/kg), followed by SPX-13, which was detected in 44% of the samples, up to 28.9 µg/kg. PTX-2, AZA-2, and PnTX-G were found in lower levels—37% of the samples showed PTX-2 (0.7–2.9 µg/kg), 29% showed AZA-2 (0.1–1.8 µg/kg), and 12% showed PnTX-G (0.4–0.9 µg/kg). To our knowledge, this was the first time that PnTX-G was observed in Galician mussels. Comparing farm places, mussels from the Ares-Sada were less affected by the lipophilic marine toxins, since neither AZA-2 nor PnTX-G were in this Ría, and only 2 samples showed low SPX-13 levels (2.6 µg/kg and 4.4 µg/kg). PTX-2, OA and DTX-2 amounts were more homogeneous, although the higher levels were found in the mussels from Ría Arousa—214 µg/kg of OA, 33.5 µg/kg of DTX-2, and 2.9 µg/kg of PTX-2. Therefore, OA and DTX-2, either free or esterified, are the major causes for concern in Galician mollusks and the higher levels were found in the larger-sized mussels, compared to the smaller ones. Despite this, no potential risk through mussel ingestion was found for the emerging toxins (SPX-13 and PnTX-G). The presence of new analogs are issues that must be considered in the shellfish safety monitoring programs through LC–MS/MS methods. Despite the data on PnTXs occurrence obtained in this study, it could not be concluded that this toxin represent a new risk for public health in Galicia. Therefore, it is necessary to monitor PnTXs to check future risks derived from mussel consumption.

## 4. Materials and Methods

### 4.1. Market Mussel Collection and Sample Preparation

A total of 41 samples of mussels (*Mytilus galloprovincialis*) were purchased weekly in five local markets and shopping centers in Lugo (Spain), during December 2018 and January 2019. A number of samples and harvest and purchase data are described in Table 3. Mussels were obtained from five commercial brands (named A, B, C, D, and E) and belonged to three Galician Rías (Ares-Sada, Arousa, and Pontevedra). Twenty-two samples were from small mussels, with about 30–35 pieces in 1 kg, and 19 samples were of a larger size, with about 24–27 individuals per kg. They were acquired fresh and alive, kept in polyethylene bags, and transported to the laboratory immediately. Afterwards, mussel meat were removed from the shell by separating the adductor muscles and the tissues connecting at hinge. After removal from the shellfish, the tissues were drained in a strainer to remove the salt water. Then, the mussels were homogenized (Ultra Turrax^TM^), kept in bags, and stored at −20 °C, protected from oxygen and light, until the analysis was carried out. Each sample was a homogenate of the soft tissue of 14–30 individual mussels (100–200 g).

### 4.2. Chemicals

Acetonitrile, methanol, sodium hydroxide, and hydrochloric acid was obtained from Panreac (Barcelona, Spain). Formic acid was purchased from Merck (Darmstadt, Germany) and ammonium formate was from Sigma-Aldrich (Madrid, Spain). All solvents were HPLC or analytical grade, and water was obtained from a water purification system (Milli-Q, Millipore, Spain). Certified reference materials were provided by Cifga (Lugo, Spain)—Dinophysistoxin-1 sodium salt (DTX1 8.08 ± 0.41 µg/g), Dinophysistoxin-2 sodium salt (DTX2 2.54 ± 0.14 µg/g), Okadaic acid sodium salt (OA 20.2 ± 1.0 µg/g), Yessotoxin (YTX 7.42 ± 0.49 μg/g), 1a-homoyessotoxin (homo-YTX 7.68 ± 0.44 μg/g), Azaspiracid-1 (AZA1 1.36 ± 0.07μg/g), Azaspiracid-2 (AZA2 1.33 ± 0.11 μg/g), and Azaspiracid-3 (AZA3 1.30 ± 0.09 μg/g). Certified calibration solutions for Pinnatoxin-G (PnTX, 2.43 ± 0.11μg/g) and for Pectenotoxin-2 (PTX-2, 5.58 ± 0.16 μg/g) were purchased from The Institute for Marine Biosciences, National Research Council Canada (Ottawa, Ontario).

### 4.3. Toxin Extraction and Hydrolysis Procedure

Samples were analyzed following the procedure described in the EU-Harmonised Standard Operating Procedure (SOP) for determination of lipophilic marine biotoxins in mollusks through LC–MS/MS [4]. The amount of 2.00 ± 0.05 g of mussel tissue homogenate were transferred into a centrifuge tube. Then, 9 mL of methanol were added, and the sample was homogenized via vortex mixing for 3 min, at the maximum speed level. Afterwards, the samples were centrifuged (3700 rpm × 10 min) at 20 °C and the supernatant was transferred to 20 mL volumetric flash. The extraction of the residual tissue pellet was repeated with another 9 mL of methanol, using a high-speed homogenizer (T 25 digital Ultra-Turrax, IKA, Staufen Germany). After centrifugation (3700 rpm × 10 min) at 20 °C, the supernatants were combined into a final volume of 20 mL with methanol. A volume of 15 mL was concentrated to 5 mL to improve the sensitivity of the analyses. One aliquot of 500 µL was filtered through a 0.22 µm filter and then analyzed through LC–MS/MS. To detect and quantify the total content of OA and DTXs, 2.5 mL of methanolic extract was hydrolyzed with 313 µL of 2.5 M NaOH. The mixture was homogenized and heated at 76 °C for 40 min. It was then cooled to room temperature, neutralized with 313 µL of 2.5 M HCl, and homogenized in the vortex. The resulting extract was filtered through a 0.22 µm filter and then analyzed through LC–MS/MS.

### 4.4. LC–MS Analysis

Analyses were performed by a 1290 Infinity ultra-high-performance liquid chromatography system, coupled to an Agilent G6460C Triple Quadrupole mass spectrometer, equipped with an Agilent Jet Stream ESI source (Agilent Technologies, Waldbronn, Germany). The toxins were separated using a column AQUITY UPLC BEH C18 (2.1 × 100 mm, 1.7 µm, Waters) at 40 °C. Mobile phase A was water and B was acetonitrile-water (95:5), both containing 50 mM of formic acid and 2 mM ammonium formate. The gradient program with a flow rate of 0.4 mL/min was started with 30% B followed by a linear gradient to 90% B in 3 min. After a linear isocratic hold time of 1.5 min at 90% B, it was returned to the starting conditions of 30% B in 0.1 min. Finally, 30% B was kept for 1.99 min before the next injection. The samples in the autosampler were cooled to 4 °C and the injection volume was 5 µL. Source conditions were—350 °C of drying gas temperature with 8 L/min flow, nebulizer gas pressure of 45 psi (Nitrocraft NCLC/MS from Air Liquid), sheath gas temperature of 400 °C, and a flow of 11 L/min. The capillary voltage was set to 4000 V in the negative mode with a nozzle voltage of 0 V, and 3500 V in the positive mode with a nozzle voltage of 500 V. The analysis was performed using two methods in the multiple reaction monitoring (MRM) acquisition mode. MS/MS Method 1 included the transitions from the regulated toxins and SPXs and PnTXs (Table 4) and the MS/MS Method 2 included just the transitions from the most common AZAs analogues [28,29], as described below.

The MS/MS screening in the positive mode, for the AZAs analogues (Method 2) was performed by monitoring the following precursor and product ions (*m*/*z*): AZA-4, AZA-5, and AZA-56 (*m*/*z* 844.5 > 826.5 and *m*/*z* 844.5 > 808.5), AZA-6 and AZA-40 (*m*/*z* 842.5 > 824.5, *m*/*z* 842.5 > 806.5), AZA-7, AZA8, AZA-9, AZA-10, and AZA-36 (*m*/*z* 858.5 > 840.5, *m*/*z* 858.5 > 822.5), AZA-11 and AZA-12, AZA-17 (*m*/*z* 872.5 > 854.5, *m*/*z* 872.5 > 836.5), AZA-13 and AZA-59 (*m*/*z* 860.5 > 842.5, *m*/*z* 860.5 > 824.5), AZA-14 and AZA-15 (*m*/*z* 874.5 > 856.5 and *m*/*z* 874.5 > 838.5), AZA-16 and AZA-21(*m*/*z* 888.5 > 870.5 and *m*/*z* 885.5 > 852.5), AZA-18 and AZA-19 (*m*/*z* 886.5 > 868.5 and *m*/*z* 886.5 > 850.5), AZA-20 and AZA-31 (*m*/*z* 900.5 > 882.5 and *m*/*z* 900.5 > 864.5), AZA-22 and AZA-23 (*m*/*z* 902.5 > 884.5 and *m*/*z* 902.5 > 866.5), AZA 24 (*m*/*z* 916.5 > 898.5 and *m*/*z* 916.5 > 880.5), AZA-26 and AZA-27 (*m*/*z* 824.5 > 806.5 and *m*/*z* 824.5 > 878.5), AZA-28 (*m*/*z* 838.5 > 820.5, *m*/*z* 838.5 > 802.5), AZA-30 and MeAZA-1 (*m*/*z* 856.5 > 838.5, *m*/*z* 856.5 > 820.5), AZA-33 (*m*/*z* 716.5 > 698.5, *m*/*z* 716.5 > 680.5), AZA-34 and AZA-39 (*m*/*z* 816.5 > 798.5 and *m*/*z* 816.5 > 780.5), AZA-35 and AZA-38 (*m*/*z* 830.5 > 812.5 and *m*/*z* 830.5 > 794.5), AZA-37 (*m*/*z* 846.5 > 828.5, *m*/*z* 846.5 > 810.5), AZA-41 (*m*/*z* 854.5 > 836.5, *m*/*z* 854.5 > 818.5), AZA-42, AZA-54, and MeAZA-2 (*m*/*z* 870.5 > 852.5, *m*/*z* 870.5 > 834.5), AZA-43 (*m*/*z* 828.5 > 810.5 and *m*/*z* 828.5 > 792.5), AZA-55 (*m*/*z* 868.5 > 850.5, *m*/*z* 868.5 > 832.5), and AZA-57 (*m*/*z* 884.5 > 866.5 and *m*/*z* 884.5 > 848.5). The collision energy (CE) was 36 eV for the bigger ion product and 44 eV for the smaller one and the fragmentor voltage (Frag) was 219 v.

### 4.5. Method Performance, Recovery, and Matrix Correction

A quality assessment of the quantification capabilities of the method on the triple quadrupole mass spectrometer was performed. Analytical method performance evaluation was conducted for three days, according to the guidelines proposed by the Regulation (EC) 657/2002 [34] and the EU-Harmonized SOP for Lipophilic toxins [4]. The external standard calibration curves were prepared in methanol with ten levels in the range of 0.09 ng/mL to 50 ng/mL (0.09, 0.19, 0.39, 0.78, 1.56, 3.12, 6.25, 12.5, 25, and 50 ng/mL). Linearity was assessed by two parameters—correlation coefficients of the quantification curves had to be greater than 0.99 (Table 1) and the deviation of the curve slopes between sample sets had to be lower than 25% to be considered as acceptable. Sensitivity of the method was evaluated as the slope of the calibration curves and the limit of detection (LOD) and limit of quantification (LOQ). LOD were—0.1 µg/kg for OA, DTX-1, DTX-2, PTX-1, and PTX-2; 0.3 µg/kg for AZA-1, AZA-2, and AZA-3; 0.1 µg/kg for SPX-13 and PnTX-G. LOQ were—0.3 µg/kg for OA, DTX-1, DTX-2, PTX-1, and PTX-2; 0.9 µg/kg for AZA-1, AZA-2, and AZA-3; 0.4 µg/kg for SPX-13 and PnTX-G.

With the aim of evaluating the effects of the extraction procedure and of the matrix, spiked mussel homogenates were used for toxin recovery and matrix correction. For this, negative mussel homogenates for the presence of toxins after being analyzed through LC–MS/MS were employed as a blank sample. The recovery obtained in the analysis of the spiked mussel homogenate (%Rspiked) was calculated as follows—%Rspiked = [(ng/mL)CALCULATED/(ng/mL)THEORETICAL] × 100. To determine the signal suppression/enhancement (SSE) due to matrix, mussel tissues were extracted following the extraction procedures described above. Toxin standards, ten concentration levels, were diluted in toxin-free mussel extracts (blank sample) and then analyzed through LC–MS/MS methods. The slopes of the curves were used for calculating the SSE factor, according to the following equation: SSE (%) = 100 × (Slope of spiked extract curve /Slope of standards curve in solvent). If SSE value was equal to 100%, no matrix effect was observed whereas a value higher than 100% meant a positive matrix effect due to an enhancement of the ionization. If this value was less than 100%, there was a negative effect, which entailed a suppression of the signal due to ionic suppression. The corrected concentration, considering the recovery and matrix effects was calculated as follows: µg toxin/kg = (µg/kg) EXTERNAL CALIBRATION × (100/%R spiked) × (100/%SSE); where, (µg/kg) EXTERNAL CALIBRATION is the concentration calculated by the external calibration prepared in methanol with 10 levels in the range of 0.09 to 50 ng/mL.

### 4.6. Expression of Results

To express the results by the toxin group according to the European legislation (as μg equivalents/kg), the use of the Toxicity Equivalent Factors (TEFs) adopted by the Scientific Panel on Contaminants in the Food Chain of the European Food Safety Authority (EFSA) is required [35]. Therefore, after calculating the individual content of each toxin/analogue, it was multiplied by the TEF before summing the total equivalents for the respective toxin group. The TEF values were—OA (1), DTX-1 (1), DTX-2 (0.6), PTX-2 (1), PTX-1 (1), AZA-1 (1), AZA-2 (1.8), AZA-3 (1.4), YTX (1), Homo YTX (1), 45-OH YTX (1), and 45-OH-homo YTX (0.5).

## Figures and Tables

**Figure 1 toxins-11-00394-f001:**
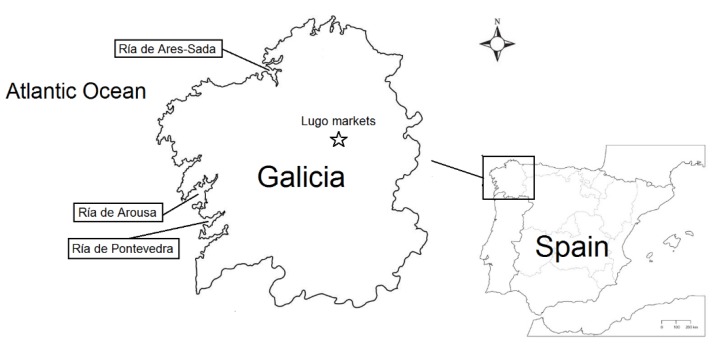
Map of Spain showing the location of Galicia (North-West Spain), mussel farms areas (Ría de Ares-Sada, Ría de Arousa, and Ría de Pontevedra) and the place from where the mussels were purchased (Lugo).

**Figure 2 toxins-11-00394-f002:**
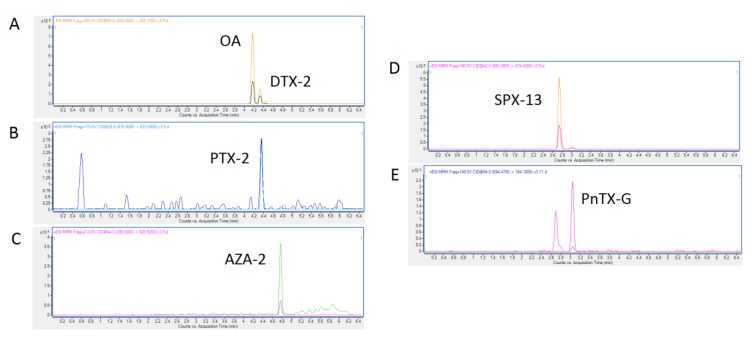
Chromatograms of okadaic acid (OA) and dinophysistoxin-2 (DTX-2) (**A**), pectenotoxin-2 (PTX-2) (**B**), Azaspiracid-2 (AZA-2) (**C**), 13-desmethyl spirolide C (SPX-13) (**D**), and pinnatoxin-G (PnTX-G) (**E**), which are found in commercial mussels (**A**–**D**) Sample No. 5; (**E**) Sample No. 11.

**Figure 3 toxins-11-00394-f003:**
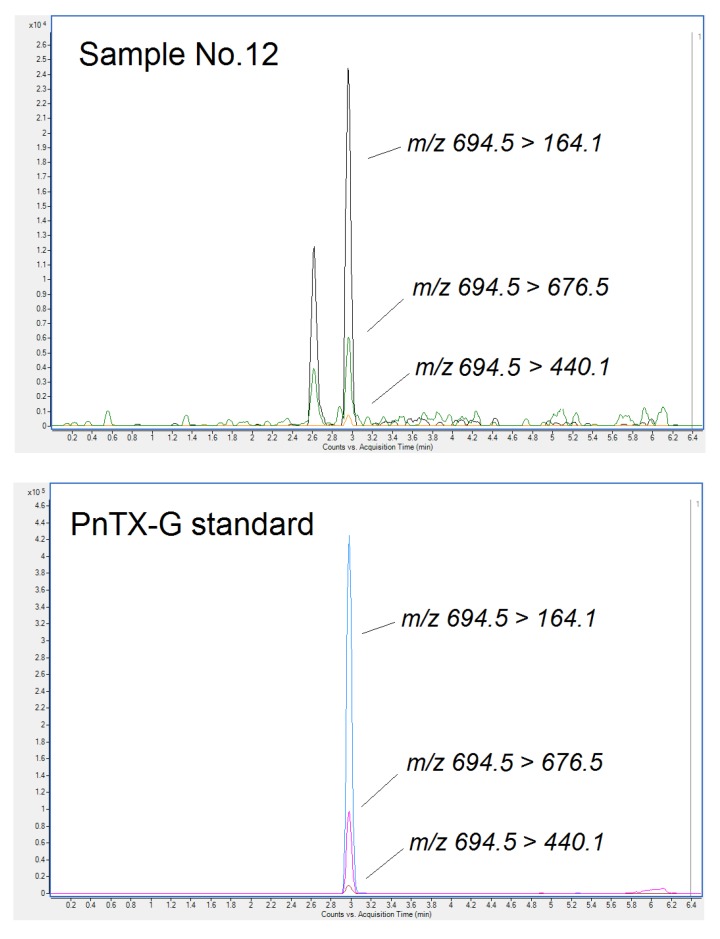
Chromatogram of specific multiple reaction monitoring (MRM) transitions for PnTX-G (*m*/*z* 694.5 > 164.1, *m*/*z* 694.5 > 440.1, and *m*/*z* 694.5 > 676.5) in Sample No. 12 and in the PnTX-G standard at a concentration of 1 ng/mL in methanol.

**Figure 4 toxins-11-00394-f004:**
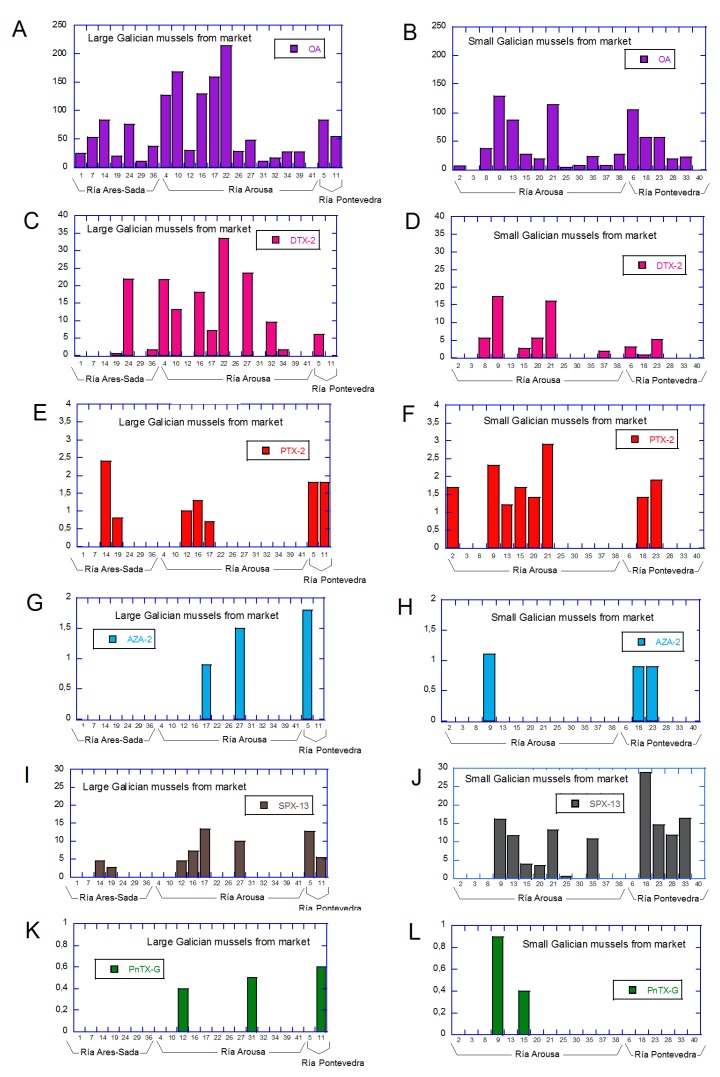
Levels of the main lipophilic marine toxins found in mussels purchased in the Lugo market (Spain). The graphs (**A**,**C**,**E**,**G**,**I**,**K**) represent the larger mussels and (**B**,**D**,**F**,**H**,**J**,**L**) represent the smaller Galicia mussels. OA (**A**,**B**), DTX-2 (**C**,**D**), PTX-2 (**E**,**F**), AZA (**G**,**H**), and PnTX-G (**I**,**J**).

**Table 1 toxins-11-00394-t001:** Calibration curves in methanol, correlation coefficients, extraction recovery (%), and signal suppression/enhancement (SSE) (%) due to matrix. Values shown are mean ± standard deviation (SD) of three experiments.

Toxin	Curves	*R* ^2^	Recovery (%)	SSE (%)
Free OA	*y* = 2381.975*x* − 561.583	0.9977	76.03 ± 15.28	211.70 ± 19.54
Free DTX-2	*y* = 2013.034*x* + 633.298	0.9937	85.24 ± 4.29	181.51 ± 6.85
PTX-2	*y* = 34362.389*x* − 5660.310	0.9987	92.34 ± 4.88	88.27 ± 6.37
AZA-2	*y* = 543036.015*x* + 83486.877	0.9962	98.8 ± 4.80	33.14 ± 1.65
SPX-13	*y* = 581714.049*x* + 292701.316	0.9997	110.48 ± 0.86	70.27 ± 6.75
PnTX-G	*y* = 387054.739*x* + 39641.076	0.9994	90.95 ± 1.61	64.74 ± 3.58
Hydrolyzed OA	*y* = 1956.583*x* − 217.441	0.9992	63. 28 ± 1.61	162.37 ± 14.85
Hydrolyzed DTX-2	*y* = 1702.876*x* + 585.842	0.9991	62.63 ± 1.63	160.25 ± 3.81

**Table 2 toxins-11-00394-t002:** Presence and concentration of lipophilic toxins in Galician mussels purchased in the market. The term ~ means below the limit of quantitation (LOQ). LOQ (OA) = 0.3 µg/kg, LOQ (DTX-2) = 0.3 µg/kg, LOQ (PTX-2) = 0.3 µg/kg, LOQ (AZA-2) = 0.9 µg/kg, LOQ (SPX-13) = 0.4 µg/kg, and LOQ (PnTX-G) = 0.4 µg/kg. Free OA and free DTX-2 means non-esterified OA and non-esterified DTX-2, respectively. Total OA is the sum of non-esterified and esterified OA. Total DTX-2 is the sum of non-esterified and esterified DTX-2. Levels of OA, PTX, and AZA group toxins are also expressed as μg OA equivalents (eq)/kg, μg PTX equivalents (eq)/kg, and μg AZA equivalents (eq)/kg, according to the Toxicity Equivalent Factor (TEF) values.

Sample No.	Free OA (µg/kg)	Total OA (µg/kg)	Free DTX-2 (µg/kg)	Total DTX-2 (µg/kg)	μg OA eq/kg	PTX-2 (µg/kg)	μg PTX eq/kg	AZA-2 (µg/kg)	μg AZA eq/kg	SPX-13 (µg/kg)	PnTX-G (µg/kg)
1	0.4	23.9	~	~	23.9	~	~	~		~	~
2	~	6.6	~	~	6.6	1.7	1.7	~		~	~
3	~	~	~	~	~	~	~	~		~	~
4	~	127.2	~	21.7	140.2	~	~	~		~	~
5	8.8	83.0	0.4	6.4	86.8	1.8	1.8	1.8	3.2	12.7	~
6	6.5	105.0	~	3.3	107.0	~	~	~		~	~
7	~	52.8	~	~	52.8	~	~	~		~	~
8	~	36.9	~	5.5	40.2	~	~	~		~	~
9	13.1	128.6	3.3	17.4	139.0	2.3	2.3	1.1	2.0	16.2	0.9
10	~	167.5	~	13.2	175.4	~	~	~		~	~
11	4.4	54.0	~	~	54.0	1.8	1.8	~		5.4	0.6
12	1.8	30.1	~	~	30.1	1.0	1.0	~		4.4	0.4
13	6.0	86.3	~	~	86.3	1.2	1.2	~		11.6	~
14	7.3	83.2	~	~	83.2	2.4	2.4	~		4.4	~
15	4.3	27.2	0.3	2.7	8.8	1.7	1.7	~		4.0	0.4
16	8.5	129.6	6.3	18.2	140.5	1.3	1.3	~		7.4	~
17	~	158.9	~	7.0	163.1	0.7	0.7	0.9	1.6	13.4	~
18	12.2	56.0	0.5	0.9	56.5	1.4	1.4	0.9	1.6	28.9	~
19	1.7	18.9	~	0.7	19.3	0.8	0.8	~		2.6	~
20	2.3	19.1	0.8	5.5	22.4	1.4	1.4	~		3.5	~
21	13.9	113.0	3.6	16.1	122.6	2.9	2.9	~		13.3	~
22	30.2	214.0	11.0	33.5	234.1	~	~	~		~	~
23	7.1	55.6	1.1	5.2	58.7	1.9	1.9	0.9	1.6	14.7	~
24	7.5	75.3	3.1	21.9	88.4	~	~	~		~	~
25	~	3.6	~	~	3.6	~	~	~		0.6	~
26	~	28.3	~	~	28.3	~	~	~		~	~
27	2.3	48.1	4.7	23.5	62.2	~	~	1.5	2.7	9.9	~
28	~	19.7	~	~	19.7	~	~	~		11.9	~
29	~	9.8	~	~	9.8	~	~	~		~	~
30	~	7.7	~	~	7.7	~	~	~		~	~
31	~	10.7	~	~	10.7	~	~	~		~	0.5
32	~	17.0	~	9.6	22.76	~	~	~		~	~
33	~	22.0	~	~	22.0	~	~	~		16.3	~
34	0.9	27.3	~	1.7	28.3	~	~	~		~	~
35	~	23.2	~	~	23.2	~	~	~		10.8	~
36	~	37.8	~	1.6	38.8	~	~	~		~	~
37	~	7.8	~	1.9	8.9	~	~	~		~	~
38	~	27.0	~	~	27.0	~	~	~		~	~
39	~	27.1	~	~	27.1	~	~	~		~	~
40	~	~	~	~	~	~	~	~		~	~
41	~	~	~	~	~	~	~	~		~	~

**Table 3 toxins-11-00394-t003:** Mussels (*Mytilus galloprovincialis*) purchased from Lugo, belonging to five commercial brands of different origins. Mussels of Brand A were collected in Ría Ares-Sada, Brands B, C, and D were collected in Ría de Arousa, and Brand E was collected in Ría de Pontevedra. The week of collection and purchase are also included.

Sample	Year	Collection and Purchase	Commercial	Mussel
No.		Week No	Brand	Size
1		49	A	Large
2		49	B	Small
3		49	C	Small
4		49	D	Large
5		49	E	Large
6		49	E	Small
7		50	A	Large
8		50	B	Small
9		50	C	Small
10	2018	50	D	Large
11		50	E	Large
12		50	B	Large
13		50	C	Small
14		51	A	Large
15		51	B	Small
16		51	C	Large
17		51	D	Large
18		51	E	Small
19		52	A	Large
20		52	B	Small
21		52	C	Small
22		52	D	Large
23		52	E	Small
24		2	A	Large
25		2	B	Small
26		2	C	Large
27		2	D	Large
28		2	E	Small
29		3	A	Large
30		3	B	Small
31	2019	3	C	Large
32		3	D	Large
33		3	E	Small
34		3	B	Large
35		3	C	Small
36		4	A	Large
37		4	B	Small
38		4	C	Small
39		4	D	Large
40		4	E	Small
41		4	B	Large

**Table 4 toxins-11-00394-t004:** MS/MS method parameters for regulated marine toxins and SPXs and PnTXs group toxins (Method 1). Precursor and product ions monitored (*m*/*z*), the fragmentor voltage (Frag), collision energy (CE), and cell accelerator voltage (CAV) in volts (V) are shown.

Toxins	Precursor Ion	Product Ion	Frag	CE	CAV	Polarity
45-OH-homo-YTX	1171.5	1091.5	250	40	4	Negative
		869.5		88		
45-OH-YTX	1157.5	1077.5	240	38	4	Negative
		871.5		86		
Homo-YTX	1155.48	1075.5	250	40	4	Negative
		869.4		88		
YTX	1141.47	1061.5	240	38	4	Negative
		855.4		86		
PTX-1	892.5	821.5	175	28	2	Positive
		213.2		44		
PTX-2	876.5	823.5	175	28	2	Positive
		213.2		44		
AZA-1	842.5	824.5	206	32	2	Positive
		806.5		44		
AZA-2	856.5	838.5	213	36	4	Positive
		820.5		44		
AZA-3	828.5	810.5	216	32	2	Positive
		792.5		44		
OA/DTX-2	803.46	113.2	350	66	7	Negative
		255.1		50		
DTX-1	817.5	255.1	350	54	7	Negative
		113		70		
SPX-13	692.45	674.4	180	42	4	Positive
		164.1		54		
SPX-13,19	678.44	660.4	149	30	4	Positive
		164.1		54		
SPX-20G	706.47	688.4	152	30	4	Positive
		164.1		54		
PnTX-G	694.47	458.3	149	30	4	Positive
		164.1		54		
PnTX-E	784.5	446.3	149	30	4	Positive
		164.1		54		
PnTX-D	782.48	446.3	149	30	4	Positive
		164.1		54		
PnTX-F	766.5	446.3	149	30	4	Positive
		164.1		54		
PnTX-B and C	741.47	458.3	149	30	4	Positive
		164.1		54		
PnTX A	712.44	458.3	149	30	4	Positive
		164.1		54		

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
