# Peer review of "LC–MS/MS Analysis of the Emerging Toxin Pinnatoxin-G and High Levels of Esterified OA Group Toxins in Galician Commercial Mussels"

_toxins, 2019, doi:10.3390/toxins11070394_

Round 1
Reviewer 1 Report
This manuscript described a lipophilic shellfish toxin profile of the Galician mussels purchased at the markets. The results of this paper are important for not only scientific society but also food safety authority and on food risk assessment.
Since the method of analysis was based on EU-Harmonized SOP for determination of lipophilic marine biotoxins, procedure should be suitable. But I think there are some considerations on the interpretation of data.
1. Although DTX3 will be hydrolyzed to OA, DTX1 and/or DTX2 (OAs), analogues esterified at carbonyl group, e.g. dinophysistoxin-4, might also be hydrolyzed to produce free form of OAs. I think it is not correct to say “confirmation of the DTX-3”.
2. Some references may not be suitable, so please cite the appropriate ones.
3. Some scientific name of the microalgae are not in italic. Please correct them.
4.“extraction” means the process of removing or obtaining something from something else (Oxford Dictionary) or the process of removing or obtaining something from something else (Longman English Dictionary). In this manuscript, “extraction” is used for possess of the sample preparation after extraction with organic solvent from mussel.
5. If you discuss about regulatory level, you should follow the regulation.
In
the case of sample 17, mentioned as exceeding regulatory level, 158.9
µg/kg of OA and 7.0 µg/kg of DTX2 were detected. Since TEF for OA and
DTX2 were 1 and 0.5, respectively, level of the OA group is 158.9*1
µg/kg + 7.0*0.5 µg/kg = 162.4 µg/kg. If the calculated level is rounded
to 2 significant digits, level of OA group is 160 µg/kg. In this case,
level of OA group in the mussel is not exceeding 160 µg/kg. In addition
this, EU regulation is set for sum of OAs and PTXs. So, please revise
the section of “Results and Discussions”. And also, explain about this
in the section of “Materials and Methods”.
My recommendations and suggestions for each part are as followings.
L10: “Mytilus galloprovincialis” should be italic.
L24: “though” should be “through” ?
L33: Although this paragraph stated general aspect of marine phycotoxins, cited reference 2 described about azaspiracids in Spanish bivalve. Please consider to delete ref. 2 or change to other reference.
L41-47: I think YTXs and PTXs are legislated only in EU. Please mention that.
L45-47:“Until 2013…..” Since it is situation in EU countries, please mention it.
In the Codex standard, no more MBA was listed and method criteria for chemical method to detect lipophilic toxins in bivalves. So, please mention about Codex standard also.
L51: Since ref 5 described only about SPX-C and analogues. I think additional references will be needed.
L58:
Ref 8 and 9 are not suitable for citing on this paragraph. Please
consider to change the references. E.g. ref 8 is described about total
synthesis
Isolation of spirolides: T. Hu, J. M. Curtis, Y. Oshima, M.
A. Quilliam, J. A. Walter, W. M. Watson-Wright and J. L. C. Wright,
Spirolides B and D, two novel macrocycles isolated from the digestive
glands of shellfish. J. Chem. Soc., Chem. Commun., 1995, 2159–2161.
Review
of spirolide: Gueret, S. M., Brimble, M. A. Spiroimine shellfish
poisoning (SSP) and the spirolide family of shellfish toxins: Isolation,
structure, biological activity and synthesis. Nat. Prod. Rep., 27, 1350-1366 (2010)
L73-:“Results and Discussion”
Please mention about all analyzed compound.
It is not able to find DTX1 was detected from sample or not. Please
make clear and I recommend showing the table in supplemental section.
And also, describe the LOQ and LOD for undetected toxins in this section and/or the section of “Materials and Methods”.
L83-84: Delete (D. acuta), (D. acuminata) and (D. caudata)
L94-: Although, it is described as “by means of the specific three transitions and the ratio between these transitions in the samples and standard”, it is impossible to find “the ratio” from Figure 3. Please consider about revising the figure.
L116: “extraction process”
Since
the standards were spiked in negative mussel “extract” (L305-L308), it
was already extracted, it is impossible to evaluate “extraction”
process. Please consider the revision.
L127: Please explain about what “Free” and “Total” mean in the legend of Table 2.
L139:
In EU, regulation or acceptable levels was set for OA group toxins as
“160 micrograms of okadaic acid equivalents per kilogram” together with
pectenotoxins, not for OA and DTX2 (there are no regulatory level for
PTX in Codex). Please consider revision.
e.g. nevertheless levels of OA group toxins were 160 µg/kg …..
L141: I think “OA group toxin” is including “DTX3 (7-O-Acyl derivatives)”. Please consider revision.
L143: Ref 26 is described about “DTX”. Please consider revision.
L161: Fig 3 should be Fig 4
L181: If you discus about the regulatory level, Table 2 should be revised following the regulatory manner. E.g. insert a column for “level of OA group toxins” deduced with TEF value of each toxins.
L197: “OA and DTX2”
It should be discuss as OA group.
L210: Although PnTXs were detected from mussels, you can’t state about necessarily for revision of legislation without risk assessment, I think. Even compound named as “toxin” is containing in the mussel, no need to set regulatory level if the “toxin” is not toxic for human consumption. I agree with necessarily of monitoring of PnTXs to evaluate the risk. Please consider revision.
L234: Is Sigma-Aldrich a distributer? If so, it should be deleted.
L236-241: I think the certified concentrations of CRMs purchased from Cifga are expressed as Na salts of these toxins. Please make sure that the calculations of each toxin were converted from Na salt to protonated form, if in needed.
L242: Insert “Canada” and name of the city after “Council”.
L264: Delete “100%”
L301: Please show the example of the levels.
L305-306: Please consider revision for “extraction”
References: If the cited regulations and SOP (e.g. ref4) are available via internet, please describe URL.
Author Response
Reviewer 1:
This manuscript described a lipophilic shellfish toxin profile of the Galician mussels purchased at the markets. The results of this paper are important for not only scientific society but also food safety authority and on food risk assessment.
Since the method of analysis was based on EU-Harmonized SOP for determination of lipophilic marine biotoxins, procedure should be suitable. But I think there are some considerations on the interpretation of data.
Question 1 (Q1). Although DTX3 will be hydrolyzed to OA, DTX1 and/or DTX2 (OAs), analogues esterified at carbonyl group, e.g. dinophysistoxin-4, might also be hydrolyzed to produce free form of OAs. I think it is not correct to say “confirmation of the DTX-3”.
Our Answer (OA1): The term DTX-3 was removed from the title.
Q2. Some references may not be suitable, so please cite the appropriate ones.
OA2: Correct references were added and wrong references were deleted.
Q3. Some scientific name of the microalgae are not in italic. Please correct them.
OA3: Scientific name of the microalgae are now in italic.
Q4.“extraction” means the process of removing or obtaining something from something else (Oxford Dictionary) or the process of removing or obtaining something from something else (Longman English Dictionary). In this manuscript, “extraction” is used for possessing of the sample preparation after extraction with organic solvent from mussel.
OA4: The terms “extracted” and “extract” was corrected.
Q5. If you discuss about regulatory level, you should follow the regulation.
In the case of sample 17, mentioned as exceeding regulatory level, 158.9 µg/kg of OA and 7.0 µg/kg of DTX2 were detected. Since TEF for OA and DTX2 were 1 and 0.5, respectively, level of the OA group is 158.9*1 µg/kg + 7.0*0.5 µg/kg = 162.4 µg/kg. If the calculated level is rounded to 2 significant digits, level of OA group is 160 µg/kg. In this case, level of OA group in the mussel is not exceeding 160 µg/kg. In addition this, EU regulation is set for sum of OAs and PTXs. So, please revise the section of “Results and Discussions”. And also, explain about this in the section of “Materials and Methods”.
OA5: The reviewer is right. To be able to compare legal levels we should include TEF values. In the new version of the manuscript, the explanaions of TEF values was included in Material and methods section (4.6. Expresion of results). In the introduction, we have already mentioned that “levels found in shellfish for human consumption must be lower than 3.75 mg eq YTX/kg, 0.16 mg eq AZA and 0.16 mg eq OA/Kg (for OA and PTX toxin group)”. According to EFSA, TEF for DTX-2 is 0.6 instead of 0.5. In the new version of the manuscript, it was included 3 new columns in table 2 to show the µg equivalents/kg for each group toxin. Also, it was explained in Results and Discussion section.
My recommendations and suggestions for each part are as followings.
L10: “Mytilus galloprovincialis” should be italic.
OA L10: Mytilus galloprovincialis is now in italic.
L24: “though” should be “through” ?
OA L24: The term was corrected.
L33: Although this paragraph stated general aspect of marine phycotoxins, cited reference 2 described about azaspiracids in Spanish bivalve. Please consider to delete ref. 2 or change to other reference.
OA L33: Reference 2 was deleted.
L41-47: I think YTXs and PTXs are legislated only in EU. Please mention that.
OA L41-47. The term Europe was included.
L45-47: “Until 2013…..” Since it is situation in EU countries, please mention it. In the Codex standard, no more MBA was listed and method criteria for chemical method to detect lipophilic toxins in bivalves. So, please mention about Codex standard also.
OA L45-47: These terms are now mentioned in the new version of the manuscript and the Codex standard was also included in the introduction
L51: Since ref 5 described only about SPX-C and analogues. I think additional references will be needed.
OA L51: Two additional references were added.
L58: Ref 8 and 9 are not suitable for citing on this paragraph. Please consider to change the references. E.g. ref 8 is described about total synthesis
Isolation of spirolides: T. Hu, J. M. Curtis, Y. Oshima, M. A. Quilliam, J. A. Walter, W. M. Watson-Wright and J. L. C. Wright, Spirolides B and D, two novel macrocycles isolated from the digestive glands of shellfish. J. Chem. Soc., Chem. Commun., 1995, 2159–2161.
Review of spirolide: Gueret, S. M., Brimble, M. A. Spiroimine shellfish poisoning (SSP) and the spirolide family of shellfish toxins: Isolation, structure, biological activity and synthesis. Nat. Prod. Rep., 27, 1350-1366 (2010).
OA L58: thank you about that. Reference number 8 was deleted and two new references were included.
L73-:“Results and Discussion ”Please mention about all analyzed compound. It is not able to find DTX1 was detected from sample or not. Please make clear and I recommend showing the table in supplemental section. And also, describe the LOQ and LOD for undetected toxins in this section and/or the section of “Materials and Methods”.
OA L73: Thank you for the comment. In the first paragraph of Results and Discussion we mention the analysed toxins. We analysed all the lipophilic toxins currently legislated in the EU and emerging toxins such as SPXs (SPX-13, SPX 13,19 and SPX-20) and PnTXs (PnTX-A, B, C, D, E, F and G). All analysed compounds are also included in Material and method section. In the manuscript we mention the toxins found in samples and DTX-1 was not in any sample. Finally, the LODs and LOQs are now included in Material and Methods section in the new version of the manuscript.
L83-84: Delete (D. acuta), (D. acuminata) and (D. caudata).
OA L83-84: Terms were deleted.
L94-: Although, it is described as “by means of the specific three transitions and the ratio between these transitions in the samples and standard”, it is impossible to find “the ratio” from Figure 3. Please consider about revising the figure.
OA L94. Figure was replaced by another one which show the ratios between transtions.
L116: “extraction process” Since the standards were spiked in negative mussel “extract” (L305-L308), it was already extracted, it is impossible to evaluate “extraction” process. Please consider the revision.
OA L127: No, it was not in this way. It was possible to evaluate the extraction process because we use mussel homogenate without being extracted. The term “extract” was wrong expressed in the manuscript, we employed this term instead homogenate. The term “mussel extract” was replaced by “mussel homogenate”. Thank you about that.
L127: Please explain about what “Free” and “Total” mean in the legend of Table 2.
OA L127: Terms “Free” and “Total” are explained in the legend of Table 2.
L139: In EU, regulation or acceptable levels was set for OA group toxins as “160 micrograms of okadaic acid equivalents per kilogram” together with pectenotoxins, not for OA and DTX2 (there are no regulatory level for PTX in Codex). Please consider revision.
e.g. nevertheless levels of OA group toxins were 160 µg/kg …..
OA L139: Sentence was rewritten.
L141: I think “OA group toxin” is including “DTX3 (7-O-Acyl derivatives)”. Please consider revision.
OA L141. The term was corrected.
L143: Ref 26 is described about “DTX”. Please consider revision.
OA L143: Two new references were added.
L161: Fig 3 should be Fig 4.
OAL161. Term was corrected.
L181: If you discus about the regulatory level, Table 2 should be revised following the regulatory manner. E.g. insert a column for “level of OA group toxins” deduced with TEF value of each toxins.
OAL181. Thank you for the suggestion. Three new columns were inserted with the amount micrograms equivalents per kg in OA, PTX and AZA group toxins.
L197: “OA and DTX2” It should be discuss as OA group.
OA L197: OA and DTX2 was replaced by toxins from OA group.
L210: Although PnTXs were detected from mussels, you can’t state about necessarily for revision of legislation without risk assessment, I think. Even compound named as “toxin” is containing in the mussel, no need to set regulatory level if the “toxin” is not toxic for human consumption. I agree with necessarily of monitoring of PnTXs to evaluate the risk. Please consider revision.
OA L210. Sentence was rewritten.
L234: Is Sigma-Aldrich a distributer? If so, it should be deleted.
OAL234: No, it is a Life Science and High Technology company. The terms is now properly included in material and methods section.
L236-241: I think the certified concentrations of CRMs purchased from Cifga are expressed as Na salts of these toxins. Please make sure that the calculations of each toxin were converted from Na salt to protonated form, if in needed.
OA L236-241: We know it but thank you about that. To avoid misunderstanding the term “sodium salt” was included so that toxins are indicated like dinophysistoxin-1 sodium salt, dinophysistoxin-2 sodium salt and OA sodium salt.
L242: Insert “Canada” and name of the city after “Council”.
OA L242: Canada and Ottawa were inserted.
L264: Delete “100%”.
OA L264: 100% was deleted.
L301: Please show the example of the levels.
OA L301. Calibration levels were included.
References: If the cited regulations and SOP (e.g. ref4) are available via internet, please describe URL.
OA: The URLs were included.

Reviewer 2 Report
This is a thorough paper and in the main well written. The more substantial comments I have are:
Table 1 is very busy and hard to read because of the over-use of<LOQ. I suggest this be replaced by a simple symbol (and referenced in the legend. At the moment, the numbers (which are of most interest) are very difficult for the reader to see.
Can lines 279 to 295 be presented another way? At the moment, it is almost impossible to read.
Is it possible to show any of the QA/QC results (described in lines 301-305)? This would add even more weight to the quality and thoroughness of the work and help practitioners appreciate the analytical standards they will need to achieve.
Less substantive comments (unclear english/typos) are:
Present not presented in line 49
Line 51, sentence starting "Nowadays, ..." is not structured properly
Line 58, sentence starting "From then..." is not structured properly
Line 60, "frequent in the" is an incorrect word usage
Line 68, what do the authors mean by "harvest furtively"?
In legend of figure 2 (they do not reference figures D and E when they name the tocxin like they have for figures A, B and C
Line 125, "an enhance of signal" is incorrect word usage
In Table 1, I am not sure if 3-4 decimal places is warranted
Line 172, correct "stablish"
Line 221/222 - "mussel meat" rather than "meat mussels"
Line 246 - "were weight" is incorrect word usage
Line 253, sensitivity, not sensibility
Line 256 - "to room" rather than "at room"
Line 260 "Analyses were" or "Analysis was"
References 14, 18 and 26 have each word in the titles capitalsied, unlike all other refernces
Author Response
Is it possible to show any of the QA/QC results (described in lines 301-305)? This would add even more weight to the quality and thoroughness of the work and help practitioners appreciate the analytical standards they will need to achieve.
OA: Yes, the correlation coefficients of the quantification curve for each toxin (linearity) are included in table 1. The LOQ and LOD for each toxin are now included in the new version of the manuscript.
Less substantive comments (unclear english/typos) are:
Line49: Present not presented in line 49
OA Line 49: The term was replaced.
Line 51, sentence starting "Nowadays, ..." is not structured properly
OA Line 51: Sentence was rewritten.
Line 58, sentence starting "From then..." is not structured properly.
OA Line 58: Sentence was rewritten.
Line 60, "frequent in the" is an incorrect word usage
OA Line 58: Sentence was rewritten.
Line 68, what do the authors mean by "harvest furtively"?
OA Line 68: Sentence was rewritten.
In legend of figure 2 (they do not reference figures D and E when they name the tocxin like they have for figures A, B and C
OA legend of figure 2: The capital letters D and E were included.
Line 125, "an enhance of signal" is incorrect word usage
OA Line 125: terms were corrected.
In Table 1, I am not sure if 3-4 decimal places is warranted
OA: yes, it is warranted. The software provided correlation coefficients of the quantification curve with 8 decimals and we rounded this value to 4 decimals.
Line 172, correct "stablish"
OA Line 172: the word “stablish” was corrected.
Line 221/222 - "mussel meat" rather than "meat mussels".
OA Line 221/222: The terms were corrected.
Line 246 - "were weight" is incorrect word usage
OA Line 246: the term was corrected.
Line 253, sensitivity, not sensibility
OA Line 253: The term was replaced.
Line 256 - "to room" rather than "at room".
OA Line 256: The term was replaced.
Line 260 "Analyses were" or "Analysis was"
OA Line 260: Term analysis was replaced by analyses
References 14, 18 and 26 have each word in the titles capitalsied, unlike all other refernces
References were corrected.

Round 2
Reviewer 1 Report
I just finished reviewing the manuscript.
The manuscript became much easier to understand.
My comment is as followings
The authors used "sodium salts" of OA, DTX-1 and DTX-2 as reference
material and they might make a calibration curves with concentrations of
the "sodium salts".
When they quantify the concentrations of OA group toxins (OAs), they
should covert the "sodium salts of OAs" to "OA", since sodium salts
containing "Na".
If the authors didn't convert the concentrations from "Sodium salts of
OAs" to "OAs", the quantified revel should be over estimate.
Please make it sure, because it's very important points in this manuscript.
Although, my English is not enough to evaluate, some parts seem to need
English correction.
Author Response
Reviewer 1:
I just finished reviewing the manuscript. The manuscript became much easier to understand.
My comment is as followings
Question 1 (Q1): The authors used "sodium salts" of OA, DTX-1 and DTX-2 as reference material and they might make a calibration curves with concentrations of the "sodium salts".
When they quantify the concentrations of OA group toxins (OAs), they should covert the "sodium salts of OAs" to "OA", since sodium salts containing "Na". If the authors didn't convert the concentrations from "Sodium salts of OAs" to "OAs", the quantified revel should be over estimate. Please make it sure, because it's very important points in this manuscript.
Our answer 1 (OA 1): thank you for the comment and we can understand the worries about this issue, but this is not a matter of concern since we did the conversion for three toxins (OA, DTX-1 and DTX-2) in the beginning of the experiments following CIFGA specifications. For example, for OA, we had into account the molecular weight of OA sodium salt which is 826.985 g/mol and the molecular weight of OA which is 805.015 g/mol. The referee is right about if we didn't convert the concentrations from "Sodium salts of OAs" to "OAs", the quantified levels should be around 2.6% over estimate. But it is not the case.
Q2: Although, my English is not enough to evaluate, some parts seem to need English correction.
AO 2: We have carefully reviewed the manuscript. Despite we have not found any significant English language mistakes to correct, we have re-write some spelling mistakes and terms. They are green underlined.